# OpenReview forum: "Numerical Sensitivity and Robustness: Exploring the Flaws of Mathematical Reasoning in Large Language Models"
_ICLR.cc/2026/Conference — Submitted to ICLR 2026_

### Official Review · Reviewer_RcEa · 2025-10-21

**Soundness:** 1
**Presentation:** 3
**Contribution:** 1
**Rating:** 2
**Confidence:** 5

**Summary:**

This paper proposes to perturb GSM8K and AIME25 from three perspectives:
1. Add irrelevant sentences without numbers.
2. Add irrelevant sentences with numbers.
3. Remove some question instructions.
The authors claim that the performance of LLMs on perturbed datasets decreases more significantly when problems are perturbed with numbers than without numbers.
The decline trend is more obvious when the perturbation intensity is larger.
They also find that LLMs can solve part of the problems whose core problem instructions are removed.

**Strengths:**

The perspective to check perturbation with and without numbers, in some sense, is interesting.

**Weaknesses:**

1. The novelty is a huge issue: The idea of inserting irrelevant context is very similar to GSM-IC. Additionally, the idea of removing the core question instruction is exactly the same as GSM-symbolic and GSM-Plus. The idea of increasing perturbation intensity is similar to E-GSM (which is not discussed in this paper).

2. Many important details about the dataset construction are not revealed, e.g.,
+ In L205-206, how do you conduct the so-called "detailed inspection"? How many annotators are involved? What is their background? Are they well-trained or not? What is the inter-agreement score among different annotators?
+ In 203-L209, the authors mention that they use GPT-4o to construct a fixed set of candidate sentences. What prompts did the authors use?
+ In L210-L215, how do you make sure: inserting sentences from another question will not change the final answer of the original question? It might be possible that the answers are changed and become wrong.

3. The evaluation setting is not revealed, which is very important for reproducibility: In L262-L265, what is the maximum response length, and have you ever checked the clip ratio for reasoning models? For AIME25, how many times do you sample to report the accuracy?

4. In Section 4.2.3, you claim that the models rely on memorization to solve problems. I believe this is more related to test set contamination, and I suggest the authors go deeper in this direction.

5. In 203-L209, the added sentences are from a fixed set, and the candidate set is relatively small. Compared to the type "with numbers", the choices to insert are smaller. This might influence the conclusion that the performance is more likely to decline for perturbation with numbers than without numbers.

6. The analysis throughout this paper is very superficial:
+ In Figure 4, the authors report the average accuracy across the whole dataset. However, in my opinion, the authors should check the question-level influence when increasing the perturbation intensity.
+ What are the intrinsic reasons why LLMs have such a phenomenon under the perturbation?
+ How to solve this kind of instability facing perturbations?

**Questions:**

See Weakness

---

### Official Review · Reviewer_d6tC · 2025-10-27

**Soundness:** 2
**Presentation:** 2
**Contribution:** 2
**Rating:** 2
**Confidence:** 4

**Summary:**

This paper investigates the robustness of large language models (LLMs) in mathematical reasoning through a sentence-level perturbation framework. The authors introduce semantically irrelevant perturbation sentences, both with numbers and without numbers, inserted into math problems from GSM8K and AIME25 benchmarks at progressively higher “perturbation intensities.” They also design a “core questioning instruction missing” variant to test whether LLMs rely on genuine reasoning or memorization.

Experiments across multiple open-source and commercial LLMs (Qwen2, DeepSeek, GPT-4o, Gemini, LLaMA, Gemma) show that models are highly sensitive to numerical perturbations, while purely semantic perturbations cause smaller but still notable degradation. The authors conclude that current LLMs may rely on pattern matching and memory rather than logical mathematical reasoning, and that perturbations reveal the limits of “robust reasoning.

**Strengths:**

- Extensive evaluation across 13 models, two benchmarks, and multiple perturbation levels. Results are reproducible and statistically clear.
- Highlights a critical limitation—numerical distractibility—that impacts real-world deployment (e.g., in finance or science). The “memorization over reasoning” finding aligns with broader concerns in the field.

**Weaknesses:**

- Similar “distractor sentence” or “context perturbation” methodologies already exist (Shi et al. 2023; Mirzadeh et al. 2024; Huang et al. 2025). The work does not propose a fundamentally new paradigm, only a modest variation.
- The paper claims to “guarantee that all factual statements are semantically irrelevant to the training samples” (lines 203–206). However, this assumption is not verifiable nor logically guaranteed under the proposed generation procedure.
The authors use GPT-4 to generate “20 factual statements semantically irrelevant to the test set” and then randomly insert them into problems. This ensures irrelevance with respect to a single question instance, but not with respect to the broader training or evaluation corpus (e.g., GSM8K, AIME25). Since the factual sentences are drawn from open-domain factual content (math, history, physics, etc.), many of them may still share semantic overlap or topical tokens (numbers, quantities, or entities) with other problems in the benchmark or even with pretraining data. Thus, it is impossible to guarantee that the perturbations are truly “semantically irrelevant” globally. More details are needed to prove your statement.
- The two perturbation types (“with numbers” vs. “without numbers”) are constructed using two entirely different pipelines: 1, Non-numeric perturbations are GPT-generated factual sentences; 2, Numeric perturbations are sampled from other GSM8K problems (by taking sentences containing numbers). This design introduces uncontrolled confounding variables — the sentence sources, styles, syntax, and lexical distributions differ across these two groups. Therefore, when the paper reports that “performance drops much more sharply with perturbations containing numbers” (Section 4.2.1, Figure 3), it is unclear whether the performance gap truly reflects numerical sensitivity or simply dataset bias. Because the “with-numbers” perturbations are drawn from the same benchmark dataset, they likely carry GSM8K-like phrasing, mathematical entities, and structural cues that overlap with real math problems — effectively producing domain-in-distribution noise, while the GPT-generated sentences are out-of-domain semantic noise. This asymmetry makes the two groups incomparable and invalidates the claim that the only difference lies in “presence of numbers.”
- Numerous typos and grammatical errors reduce professionalism and clarity (e.g., lines 14 true the, 101 without containing numbers, 179–180). A thorough proofread is essential.

## References
- Shi et al. 2023, Large Language Models Can Be Easily Distracted by Irrelevant Context
- Mirzadeh et al. 2024, GSM‑Symbolic: Understanding the Limitations of Mathematical Reasoning in Large Language Models
- Huang et al. 2025, MATH‑Perturb: Benchmarking LLMs’ Math Reasoning Abilities against Hard Perturbations

**Questions:**

see weaknesses

---

### Official Review · Reviewer_jDUk · 2025-11-01

**Soundness:** 2
**Presentation:** 3
**Contribution:** 2
**Rating:** 2
**Confidence:** 4

**Summary:**

This paper investigates LLM's robustness to context-irrelevant perturbations in queries. The authors perturb the original query via inserting semantic-irrelevant sentences with or without numbers, and by varying the number of perturbation instances per query. They found that LLMs are more sensitive to numerical perturbations, and performances decline further as number of perturbation increases. They also tried removing the final core question statement in the GSM8K dataset, and found that LLMs tend to find the answers even without being instructed, suggesting memorization or data contamination.

**Strengths:**

1. The paper is well written and easy to follow.
2. The experiments are straight forward and covers a variety of LLMs.
3. The core question removal is an interesting discovery.

**Weaknesses:**

1. While the authors claim to propose a novel perturbation method, similar approaches have been explored in the literature. For example, GSM-IC [1] (which you referenced in line 80) found that LLMs can be distracted by irrelevant context, which directly contradicts your statement *"...still remain open that whether semantically irrelevant perturbations can affect the problem solution"*. Coleg [2] also investigated extending question context length with (mostly) irrelevant details, and proposed a training strategy to increase model's robustness to such perturbations.
2. The authors only exposed the LLM's weakness to irrelevant perturbations, but did not propose an effective way to resolve such weakness. This finding alone lack novelty and contribution, since it was already explored in the literature.
3. The experiments are not comprehensive enough. From my understanding, the reason LLMs tend to be distracted by irrelevant context, is that they are explicitly trained to utilize all available informations in the query during instruction tuning. Based on this hypothesis, it's better if the authors directly prompt the LLMs to ignore irrelevant context, and use their performance in this setting as an additional baseline.

[1] [Large Language Models Can Be Easily Distracted by Irrelevant Context](https://arxiv.org/abs/2302.00093)

[2] [Can LLMs Solve longer Math Word Problems Better?](https://arxiv.org/abs/2405.14804)

**Questions:**

1. Since the perturbations are all semantic-irrelevant, would it be more appropriate to classify LLM's weakness as poor question understanding / condition extraction rather than flawed reasoning / numerical sensitivity?
2. Would LLMs be more robust to such perturbations when prompted to ignore irrelevant context?
3. What's the contribution of this paper, given that previous works have already studied the effects of irrelevant perturbations?

---

### Official Review · Reviewer_vKWg · 2025-11-01

**Soundness:** 2
**Presentation:** 3
**Contribution:** 2
**Rating:** 2
**Confidence:** 4

**Summary:**

This paper investigates the numerical sensitivity and robustness of large language models (LLMs) in mathematical reasoning tasks.
The authors observe that models often fail when the problem statement is perturbed with irrelevant sentences containing numbers, even if those numbers are unrelated to the question. To systematically study this, they propose a new evaluation framework that injects irrelevant sentences—either numerical or non-numerical—at varying levels of intensity into GSM8K and AIME problems. Additionally, they introduce a “missing question instruction” setup to test whether models rely on memorization or true reasoning. Experiments reveal that most models, especially open-source ones, experience substantial accuracy drops (up to 50%) under numerical perturbations, while closed-source reasoning models (e.g., GPT-4o, o3, Gemini) are more robust but still affected. The results highlight that current LLMs lack the ability to filter out irrelevant numerical information and rely heavily on superficial pattern matching rather than genuine reasoning.

**Strengths:**

1. The paper systematically explores multiple perturbation types (numerical vs. non-numerical) and intensities, scaling from single-sentence insertions to twice the original problem length. The inclusion of both GSM8K and AIMEprovides a well-rounded evaluation across difficulty levels.

2. Wide model coverage – The study evaluates a broad spectrum of models, including open-source LLMs (Qwen, DeepSeek, LLaMA, Gemma) and proprietary reasoning models, offering valuable cross-model insights into robustness differences between architectures and training paradigms.

**Weaknesses:**

1. The core finding that models are brittle to irrelevant or noisy context—has been partially demonstrated in prior works such as GSM-Plus and MathCheck. The paper could benefit from a deeper discussion on what unique insight it contributes beyond confirming existing robustness issues, or how its new findings could be applied.

2.Several promising analyses are missing: (1) The perturbation length is relatively small. Scaling the irrelevant text by 10× or 100× could better approximate real-world long-context reasoning and reveal how degradation behaves across positions (problem placed at the beginning, middle, or end of the context). (2) It would be valuable to test whether simple prompting strategies (e.g., instructing the model to ignore irrelevant numbers) can mitigate this vulnerability. (3) Conducting fine-tuning or robustness training on the proposed perturbed data could show whether these weaknesses can be systematically reduced, thus validating the framework’s utility beyond evaluation.

**Questions:**

1. Have the authors tested longer-context scenarios (e.g., 10×–100× text length) to see whether accuracy degradation continues linearly or plateaus?

2. Can explicit prompt instructions (such as “use only numbers directly related to the question”) reduce the negative effect of numerical noise?

3. Would training or fine-tuning on the proposed perturbed datasets improve model robustness against irrelevant information?

---

### Meta-Review · Area_Chair_E3cw · 2025-12-28

**Summary:**

Paper Summary. This paper studies sensitivity and robustness of LLMs in mathematical reasoning tasks with respect to context-irrelevant perturbations. The injects semantic-irrelevant sentences, including numerical or non-numerical, into the original question statements. Experiments show that LLMs are more sensitive to numerical perturbations with substantial accuracy drops, and performances decline further with more perturbations. They also introduce a “missing question instruction” setup by removing the final core question statement, and found that LLMs still find the answers even without being instructed, which indicates LLMs rely on memorization rather than true reasoning.

Paper Strengths. (1) The paper highlights a critical limitation of LLM's robustness to numerical perturbations. (2) The paper systematically covers multiple perturbation types. (3) It also includes a broad coverage of models.

Reviewer Concerns. The major concerns from reviewers is that (1) The finding of model brittleness to context-irrelevant perturbations is not new, which has already been explored in previous work. (Reviewer vKWg, Reviewer jDUk, Reviewer d6tC, Reviewer RcEa). (2) The perturbation generation algorithm is not entirely convincing (Reviewer d6tC, Reviewer RcEa). There is no verification of perturbation to be semantically irrelevant. It is also unclear that the performance drop truly reflects numerical sensitivity or other uncontrolled confounding variables. (3) The analyses is weak from multiple perspectives. The perturbation length is relatively small and does not include position analysis, which is important for real-world long-context reasoning. (Reviewer vKWg, Reviewer RcEa). (4) The paper does not include mitigation method exploration such as prompting engineering and robustness fine-tuning. (Reviewer vKWg, Reviewer jDUk).

**Reviewer Concerns:**

The author does not provide response, so the reviewer concerns have not been addressed.

**Reviewer Scores:**

The author does not provide response, so I would expect the reviewer score remains the same. Rating: 2/2/2/2. Confidence: 4/4/4/5.

---

### Decision · Program_Chairs · 2026-01-26

Reject